# Versatile and Robust Method for Antibody Conjugation to Nanoparticles with High Targeting Efficiency

**DOI:** 10.3390/pharmaceutics13122153

**Published:** 2021-12-14

**Authors:** Indra Van Zundert, Maria Bravo, Olivier Deschaume, Pierre Cybulski, Carmen Bartic, Johan Hofkens, Hiroshi Uji-i, Beatrice Fortuni, Susana Rocha

**Affiliations:** 1Molecular Imaging and Photonics, Department of Chemistry, KU Leuven, Celestijnenlaan 200F, 3001 Heverlee, Belgium; indra.vanzundert@kuleuven.be (I.V.Z.); maria.bravo@kuleuven.be (M.B.); pierre.cybulski@kuleuven.be (P.C.); johan.hofkens@kuleuven.be (J.H.); hiroshi.ujii@kuleuven.be (H.U.-i.); 2Soft-Matter Physics and Biophysics, Department of Physics and Astronomy, KU Leuven, Celestijnenlaan 200D, Box 2416, 3001 Heverlee, Belgium; olivier.deschaume@kuleuven.be (O.D.); carmen.bartic@kuleuven.be (C.B.); 3Research Institute for Electronic Science (RIES), Hokkaido University, N20W10, Kita Ward, Sapporo 001-0020, Japan

**Keywords:** mesoporous silica nanoparticles, antibody functionalization, targeted drug delivery systems

## Abstract

The application of antibodies in nanomedicine is now standard practice in research since it represents an innovative approach to deliver chemotherapy agents selectively to tumors. The variety of targets or markers that are overexpressed in different types of cancers results in a high demand for antibody conjugated-nanoparticles, which are versatile and easily customizable. Considering up-scaling, the synthesis of antibody-conjugated nanoparticles should be simple and highly reproducible. Here, we developed a facile coating strategy to produce antibody-conjugated nanoparticles using ‘click chemistry’ and further evaluated their selectivity towards cancer cells expressing different markers. Our approach was consistently repeated for the conjugation of antibodies against CD44 and EGFR, which are prominent cancer cell markers. The functionalized particles presented excellent cell specificity towards CD44 and EGFR overexpressing cells, respectively. Our results indicated that the developed coating method is reproducible, versatile, and non-toxic, and can be used for particle functionalization with different antibodies. This grafting strategy can be applied to a wide range of nanoparticles and will contribute to the development of future targeted drug delivery systems.

## 1. Introduction

Despite the numerous advances in treatment options, cancer remains a leading cause of mortality worldwide. Existing methods bear limitations and complications, such as incomplete removal of the tumor and severe side effects. Therefore, a combination of treatments is often required to reach the desired results [1,2,3]. The urge to develop more effective therapies gave rise to intensive research in delivery of chemotherapeutics using nanoparticles. Engineered nanoparticles have been shown to serve as excellent drug nano-carriers. Among the advantages of using nanoparticles are their higher drug-loading capacity, the protection of the drugs against degradation during blood circulation, and the possibility to easily add other functionalities. As the size of particles can be tailored, nanoparticles between 20 and 200 nm can take advantage of the enhanced permeability and retention (EPR) effect, to passively accumulate near the tumor because of abnormal blood vessel architecture [4,5]. However, over the last years, increasing debate on the EPR effect has emerged, raising doubts about its reliability and applicability [6,7,8]. Moreover, it has been repeatedly reported that only a small percentage of nanoparticles intravenously injected in mouse models actually reaches the tumor cells [9,10,11]. This is caused by the obstacles or biological barriers encountered by nanoparticles, which limit their delivery to the tumor. These include the bloodstream, the innate immune system, the endothelial wall, and the dense extracellular matrix (ECM) of the tumor [12,13]. The ECM consists of fibers (such as collagen and fibronectin), which are known to hamper the diffusion of nanoparticles significantly [14]. This, together with an increased interstitial fluid pressure at the tumor site, poses a substantial barrier for nanoparticle transport to the tumor. To this end, strategies that can enhance the delivery of nanoparticles are widely being explored today. Active targeting of nanoparticles to cancer cells is one of those strategies [15,16].

Over the years, a wide range of nanoparticles have been engineered and several approaches have been developed to promote nanoparticle internalization into specific cells. Often, nanoparticles are functionalized with ligands that recognize overexpressed receptors or markers present on the cancer cell membrane [5,17,18]. In doing so, they facilitate specific accumulation of the drug in cancer cells [19]. Folic acid or transferrin-conjugated nanoparticles are popular examples of such drug delivery systems (DDSs), as they bind to folate and transferrin receptors, respectively, overexpressed in certain cancers [18,20,21]. Typically, one nanoparticle is designed against a particular receptor or marker, hence targeting a specific cancer. However, patients with the same type of cancer can overexpress different markers. For instance, overexpression of the estrogen receptor (ER) is linked to a hormone-sensitive form of breast cancer (ER+), while HER2 is overexpressed in an aggressive and fast-growing type of breast cancer (HER2+) [22,23,24]. Given the variety in potential targets, there is a continuous search for simple methods to customize nanoparticles, turning them into versatile nano-carriers. To this end, antibodies have proven to be a promising strategy as they can be developed against most of the existing markers. The success of antibodies in targeting tumor cells has already been proven with the development of antibody–drug conjugates, which have emerged as powerful therapeutic agents in cancer therapies. To date, nine ADCs have been approved by the FDA for clinical use [25,26,27,28,29]. After the advances in antibody-drug conjugates, conjugation of antibodies to nanoparticles yields great therapeutic potential [30,31].

Conjugation of antibodies to nanoparticles can be achieved via different strategies, i.e., ionic adsorption (non-covalent attachment) [32], covalent binding (including carbodiimide chemistry [33], maleimide chemistry [34] and click-chemistry [35]) or using adapter molecules such as biotin [36]. In ionic adsorption, antibodies are linked to the nanoparticles via electrostatic interactions [37], leading to poor reproducibility and low stability [38]. Alternatively, adapter molecules, such as the avidin-biotin couple, can be implemented, but this interaction is influenced by the pH, affecting nanoparticle stability in more acidic conditions as found in the tumor microenvironment [39].

Covalent attachment is achieved by functionalizing the nanoparticle surface with functional groups (e.g., amine, carboxylic, maleimide etc.), which can react with the amino acid side chain of the antibody by standard conjugation methods. Covalent attachment of antibodies is generally preferred, provided that an appropriate approach is used. For instance, while EDC/NHS coupling is a common method for covalent attachment [40], it can result in oligomerization of antibody molecules [41]. Furthermore, some conjugation methods require the use of catalysts, often metals, that can lead to increased toxicity of the nanoparticles if not fully removed from the solution [42,43]. Therefore, when using covalent conjugation, a catalyst-free approach and vast optimization are important. Despite the progress in nanoparticle functionalization, there is still an urge for simple and reproducible strategies to conjugate antibodies to nanoparticles, enabling the development of versatile DDSs, which can easily be customized for selectivity towards different cancer markers.

In this work, we propose a simple and reproducible coating strategy for antibody conjugation to nanoparticles. To avoid the use of catalysts, we developed an approach based on copper-free click chemistry (Figure 1). Briefly, antibodies were labelled with a dibenzocyclooctyne (DBCO) moiety, while an Azide (N_3_) group was attached to the nanoparticles. The independent activation of the antibody and nanoparticles reduces the possibility of oligomerization of the antibody or aggregation of the particles. We proved the versatile nature of our method by creating two different types of particles, either conjugated to an anti-Cluster of Differentiation 44 (anti-CD44) or anti-Epidermal Growth Factor Receptor (anti-EGFR) antibody, targeting CD44 or EGFR overexpressing cells, respectively. CD44 and EGFR are surface receptors that manifest themselves as important bio-markers in cancer [44,45]. In this report, mesoporous silica nanoparticles (MSNPs) were used as a model application for our coating strategy. In recent years, MSNPs have been pointed out as extremely promising tools in cancer research given their high biocompatibility, chemical stability, high drug-loading and -releasing capacities, straightforward functionalization and low-cost, scalable fabrication [46,47,48]. For these reasons, MSNPs were chosen as a study model of DDSs for further surface modifications. Nevertheless, we foresee that the coating strategy here presented can be easily applied to a wide range of nano-carriers besides MSNPs, as it only requires the presence of amine groups on the surface of the nanoparticle. This simple conjugation strategy will contribute to the up-scaling of antibody-conjugated nanoparticles and to the future developments of targeted nanoparticles with multiple functionalizations.

## 2. Materials and Methods

### 2.1. Materials

Mouse anti-human EGFR (αEGFR, monoclonal, cat. BE0278) and rat anti-human CD44 (αCD44, monoclonal, hermes-1, cat. BE0039) were purchased from Bio X Cell (Lebanon, NH, USA). Goat anti-rabbit IgG (cat. ab6702) was purchased from Abcam (Cambridge, United Kingdom). Secondary antibodies, donkey anti-rat IgG Alexa Fluor 488 (cat. A-21208)goat anti-mouse IgG Alexa Fluor 488 (cat. A11001), Dulbecco’s modified eagle medium (DMEM), LysoTracker^TM^ Deep Red, DiR lipophilic dye, Gentamicin, Dulbecco’s phosphate buffered saline (PBS, no calcium, no magnesium), Formaldehyde (4% in PBS), trypsin-EDTA (0.5%), Hank’s balanced salt solution (HBSS, no phenol red), Ethanol (absolute, 99.9%), donkey-anti-rat IgG-Alexa Fluor 488 (cat. A-21208), and Zeba™ Spin Desalting Columns (40K MWCO, 0.5 mL) were purchased from ThermoFisher Scientific (Waltham, MA, USA). Tetraethyl orthosilicate (TEOS, 98%), cetyltrimethylammonium chloride solution (CTAC, 25% in H_2_O), triethanolamine (TEA, 99%), hydrochloric acid (HCl, 1 N), Rhodamine B basic violet 10 (RhoB, 93%), Fluorescein isothiocyanate (FITC), (3-Aminopropyl)triethoxysilane (APTES), polyethyleneimine solution (PEI, 50% *w/v* in H_2_O), Triton X-100 (0.1%), *N*-(3-dimethylaminopropyl)-*N*’-ethyl-carbodiimide (EDC, 97%), Dimethyl sulfoxide (DMSO, >99.5%) and 3D Petri Dish^®^—Microtissues were purchased from Sigma Aldrich (Saint Louis, MO, USA). Atto488 and Atto565 NHS ester conjugate were purchased from Atto-TEC GmbH (Martinshardt, Siegen, Germany). DBCO-PEG4-NHS-ester and N_3_-PEG_4_-NHS-ester were purchased from Click Chemistry Tools (Scottsdale, AZ, USA). The A549 cell line was obtained from Sigma Aldrich (ECACC 86012804). The A431, Hek293T and NIH3T3 cell lines were a kind gift from Prof. Hideaki Mizuno (KU Leuven, Belgium) [49,50]. The HepG2 cell line was obtained from Prof. Hitoshi Kasai (Institute of Multidisciplinary Research for Advanced Materials (IMRAM, Chome-1-1 Katahira, Aoba Ward, Sendai, Miyagi 980-8577, Japan [51]). BJ1-hTERT cells were a kind gift of Dr. Barderas (Instituto Salud D. Carlos III, Spain). FuGENE^®^ 6 and Alexa Fluor 488 HaloTag^®^ ligand were both purchased from Promega (Madison, WI, USA). The pcDNA3-EGFR-HaloTag^®^ plasmid was a kind gift from the laboratory of Prof. Dr. Hideaki Mizuno (KU Leuven, Belgium). All the chemicals were used without further purifications.

### 2.2. CD44, EGFR and IgG Antibody Labelling

The antibodies (rat anti-human CD44, mouse anti-human EGFR and goat anti-rabbit IgG) were functionalized with DBCO-PEG_4_-NHS ester for conjugation onto the nanoparticles via copper-free click chemistry. In addition, a fluorescent label, Atto565-NHS ester (for the EGFR antibody and the IgG) or Atto488-NHS ester (for the CD44 antibody), was sometimes grafted onto the antibody for further investigations via fluorescence microscopy. The labelling of 500 µg of antibody was carried out in 50 mM of borate buffer pH 8.5 (antibody concentration 1 mg/mL). Then, 10 molar equivalents of DBCO-PEG_4_-NHS ester and 2 molar equivalents of Atto565-NHS ester or Atto488-NHS ester (when needed) were added to the reaction, according to Eggermont and Hammink et al. [52]. After 6 h under magnetic stirring at room temperature, the dual labelled antibodies were purified over a 0.5-mL 40K Zeba desalting column to remove residual-free Atto565-NHS ester and DBCO-PEG_4_-NHS ester molecules. After purification, absorption measurements were carried out using a UV-VIS spectrophotometer (BioDrop μLite, BioChrom in Appendix A). The absorption at different wavelengths was used to determine the concentration of protein, DBCO and fluorescence dyes (A_280_ for the antibody, A_309_ for DBCO, A_488_ for Atto488 and A_565_ for Atto565). The values obtained corresponded to a labelling degree of an average of 10 molecules of DBCO and 2 dye molecules per antibody (absorption spectra shown in Appendix A). Previous reports have shown that this degree of labelling does not induce loss of antibody specificity [53].

### 2.3. Synthesis MSNPs

The MSNPs were synthetized by the biphase stratification method reported by Shen et al. [54]. In short, 0.18 g of TEA was mixed with a solution of 24 mL of CTAC and 36 mL of milli-Q water. This mixture was heated to 60 °C under magnetic stirring for 1 h. Next, 20 mL of TEOS (20 *v*/*v* % in octadecene) was slowly added with a syringe and the reaction was kept proceeding overnight. When fluorescein (FITC) was linked to the MSNPs’ matrix (FITC encapsulation), 16.6 mg of FITC was dissolved in 10 mL 99.8% ethanol and 400 μL of (3-Aminopropyl)triethoxysilane (APTES). This mixture was stirred for 2 h under inert atmosphere to couple FITC to the aminosilane. After 2 h, the solution was added together with the TEOS. Next, the reaction was cooled down to room temperature and the nanoparticles were washed with a solution of HCl 1.1 M in water/ethanol (*v/v* = 1.25:10) with centrifugation-dispersion-sonication cycles to remove CTAC from the pores.

Subsequently, the nanoparticles were washed two times with milli-Q water in order to neutralize the pH.

### 2.4. MSNP Dye/Drug Loading

The pores of the MSNPs were loaded with doxorubicin (Dox) or rhodamine B (RhoB) for cytotoxicity experiments and fluorescence imaging, respectively. Loading of RhoB was performed in milli-Q water under magnetic stirring for 3 h. For Dox loading, MSNPs were first dispersed in phosphate buffer (pH 9) to maximize the loading efficiency. To avoid Dox aggregation, the solution containing Dox and MSNPs was sonicated for 10 min. Next, the solution was stirred for 24 h at 400 rpm. After loading (of Dox or RhoB), the solution was centrifuged and the supernatant was replaced with milli-Q water and the Dox- and RhoB-loaded nanoparticles were re-suspended (MSNPs_Dox and MSNPs_RhoB, respectively). The supernatants of all the centrifugation steps were collected and measured with a spectrometer in order to quantify the Dox loaded inside the MSNPs. We estimated a Dox concentration in the MSNPs of approximately 50 µM.

### 2.5. MSNP Functionalization

To coat the nanoparticles with a PEI layer, a 0.75% PEI solution (in milli-Q water), adjusted to pH 7 (with 37% HCl) was added to the dye or drug-loaded MSNPs (1:1 ratio) in a plastic vial. This mixture was magnetically stirred for 3 h, yielding PEI-coated MSNPs. To facilitate the Azide conjugation, PEI-coated particles were dispersed in borate buffer (pH 8.5). To conjugate the NHS ester-PEG_4_-N_3_ linker to the PEI amine groups, an NHS ester reaction was used. In detail, 1.5 mg NHS ester-PEG_4_-N_3_ (in DMSO) were added to 500 µL of PEI-MSNPs (10 mg/mL) in a dropwise manner and magnetically stirred for 4 h. After the reaction, the nanoparticles were centrifuged and re-dispersed in borate buffer (N_3_-PEI-MSNPs). To conjugate the desired antibody via a copper-free click reaction, 100 µg of labelled antibody (50 µL of 1 mg/mL labelled antibody solution in borate buffer) were added to 450 µL of N_3_-PEI-MSNPs (10 mg/mL). The reaction was stirred for 6 h at room temperature. After the reaction, nanoparticles (Ab-PEI-MSNPs) were centrifuged at low speed (700 RPM) and re-dispersed in milli-Q water.

### 2.6. Ab-PEI-MSNPs’ Characterization

The synthesized nanoparticles were characterized by confocal fluorescence microscopy, scanning electron microscopy (SEM,) and atomic force microscopy (AFM). For the fluorescence microscopy experiments, 60 µL Ab-PEI-MNSP solution was pipetted in a Coverwell^TM^ perfusion chamber (ThermoFisher Scientific) placed onto a #1 coverglass. After 30 min, when some nanoparticles had sedimented, the sample was imaged with a Leica TCS SP8 mini microscope (Wetzlar, Germany). For SEM measurements, nanoparticles were drop-casted onto an Indium-Tin Oxide-coated glass and dried. Next, the glass was coated with Au/Pd for 20 s. Nanoparticles were visualized using a FEI Quanta 250 FEG Scanning Electron Microscope (ThermoFisher Scientific). Zeta potential measurements were carried out on a Malvern Zetasizer system (Malvern, UK). For AFM characterization, an Agilent 5500 AFM with MAC III controller was used for morphological imaging in intermittent contact mode in air. MSNL-F (f = 120 kHz, k = 0.6 N m^−1^, tip radius of curvature < 12 nm) probes were used. The AFM topography images were leveled, line-corrected, and measured (line profiles for diameter determination) using Gwyddion, a free and open-source SPM (scanning probe microscopy) data visualization and analysis program (version 2.48) [55]. AFM samples were prepared on silicon substrates freshly cleaned in piranha solution. For bare and functionalized MSNPs that carry a negative surface charge, the clean substrate was first incubated in PAH, followed by rinsing and drying. On the other hand, PEI-modified MSNPs were deposited on a bare silicon substrate. For each sample, the nanoparticle suspension was incubated for 1 min on the substrate, before rinsing with ultrapure water and drying with pure nitrogen gas. To check the colloidal stability of the Ab-PEI-MSNPs, turbidity measurements were performed. Accordingly, the sample turbidity (optical density) of the nanoparticles dissolved in different media (milli-Q water, FBS and DMEM+ 10% FBS) was determined before and after 24 h of incubation at 37 °C by measuring the absorbance at 600 nm.

### 2.7. Cell Culture

A549 cells, HepG2, BJ1-hTERT, NIH3T3, Hek293 and A431 cells were cultured in 25 cm^2^ culture flasks at 37 °C under 5% CO_2_ atmosphere. All cell lines were maintained in DMEM medium with 10% FBS, 1% l-glutamax and 0.1% gentamicin. For fluorescence microscopy experiments, the cells were seeded in 29-mm, glass-bottom dishes (Cellvis, Mountain View, CA, USA) and grown until~60% confluency before adding the nanoparticles.

### 2.8. Immunofluorescence Labeling

A549 and HepG2 cells were stained with both the dual-labelled (with DBCO and Atto488) and non-labelled CD44 antibody (rat anti-human CD44). First, cells were seeded in two glass-bottom dishes and grown overnight. Next, the cells were fixed with paraformaldehyde (4%) and the membrane was permeabilized with Triton X-100 (0.1%), for 10 min. The sample was carefully washed with PBS (1×) between each step. After washing, blocking was performed for 1 h with a bovine serum albumin solution (3% in PBS). The dual-labelled and non-labelled CD44 antibodies were added to the cells at a final concentration of 2 μg/mL and incubated overnight at 4 °C. After antibody incubation, the dual-labelled antibody samples were washed three times with PBS. The samples containing non-labelled antibody were washed three times with PBS and incubated with the secondary antibody, donkey-anti-rat IgG-AF488, at a final concentration of 1 μg/mL for 2 h. After that, the samples were washed with PBS. The same protocol was used for the immunofluorescent staining of A431 and Hek293 cells with both the dual-labelled and non-labelled EGFR antibody (mouse anti-human EGFR). In this case, a goat-anti-mouse IgG AF488 secondary antibody was used after incubation with the non-labelled antibody.

### 2.9. Ab-PEI-MSNPs’ Targeting Efficiency

A549 cells, HepG2, BJ1-hTERT, NIH3T3, Hek293 and A431 cells were seeded in a 29 mm, glass-bottom dish and grown until 60–80% confluency. FITC-encapsulated nanoparticles with different functionalization (bare MSNP, PEI-coated MSNPs, EGFR or CD44 antibody-conjugated MSNPs, and IgG-conjugated MSNPs) were added to the cells to a final concentration of 50 µg/mL. After 6 h of incubation with the NPs, the cells were washed three times with PBS and fresh medium was added to the samples. The samples were placed back in the incubator for an additional 24 h incubation. Prior to imaging, the plasma membrane was stained using DiR (1 µM) in HBSS for 13 min and the sample was washed three times with HBSS. The resulting images were analyzed to quantify the amount of nanoparticle internalization by calculating the mean fluorescence intensity of the nanoparticle signal inside at least 20 cells per condition using the Fiji open source software (version 1.51). [56]. Briefly, the cell area (region of interest, ROI) was manually selected based on the membrane staining. Next, the mean fluorescence intensity of the nanoparticles was calculated for each ROI.

### 2.10. Transfection of Hek293

Hek293 was reverse transfected with a pcDNA3-EGFR-HaloTag^®^ [50]. Transfection was carried out according to the supplier’s protocol (FuGENE^®^ 6, Promega). Briefly, 1 µg of DNA was added to 100 μL of serum-free DMEM medium together with 3 μL of the transfection reagent. This mixture was vortexed shortly and incubated for 17 min. Meanwhile, the cells were passaged and plated in 29-mm, glass-bottom dishes (Cellvis). After 17 min of incubation, the transfection mixture was added to the plated cells in a dropwise manner. The next day, αEGFR-PEI-MSNPs were added to the transfected Hek293 cells. After 3 h, the medium was refreshed to avoid continued uptake of nanoparticles. The cells were incubated with the nanoparticles for a total of 24 h. After incubation, EGF receptor was visualized in the transfected Hek293 cells after incubation with the Alexa Fluor 488 HaloTag^®^ ligand (Promega) for 30 min at 37 °C (final ligand concentration of 250 nM in cell medium).

### 2.11. Fluorescence Microscopy

Confocal fluorescence imaging was performed on a Leica TCS SP8 mini microscope implementing a HC PL APO 63× water immersion objective (NA 1.2). Distinct diode lasers were used depending on the dye. For LysoTracker^TM^ DeepRed and DiR lipophilic dye, a red, 638-nm diode laser was used at a laser power between 10 and 60 µW. For RhoB or Atto565 detection, a green, 552-nm diode laser was used for excitation (laser power between 20 and 80 µW), while the blue, 488-nm diode laser was used to excite Doxorubicin, FITC and Atto488 (laser power between 10 and 50 µW). The laser powers were measured at the objective. Detection was performed with HyD SMD high-sensitivity detectors in standard mode, operating in a detection range of 400 to 800 nm. The detection range was adjusted depending on the dye, with 500–550 nm for Atto488 and FITC detection, 570–600 nm for Doxorubicin, RhoB, and Atto565 and 650–750 nm for LysoTracker^TM^ DeepRed and DiR lipophilic dye detection. The images were acquired with a z-step of 1 µm and line averaging of 3.

### 2.12. The αCD44-PEI-MSNPs’ Intracellular Localization

A549 cells were incubated with αCD44-PEI-MSNPs_RhoB for 24 h (at a final concentration of 50 µg/mL). The sample was then washed three times with PBS to remove extracellular nanoparticles and kept in HBSS during image acquisition. One sample was imaged immediately after 24-h incubation, while two other replicates were placed back in the incubator for the 48-and 72-h time points (after replacing the PBS for cell culture medium). Prior to imaging, the samples were washed with HBSS and the lysosomes were stained with LysoTracker^TM^ Deep Red (100 nM final concentration in HBSS) for 15 min. After washing three times with PBS, the samples were imaged in HBSS. The fluorescence images acquired (see Fluorescence microscopy section) were processed and analyzed using Fiji and the built-in, co-localization plugin JACoP [57]. Within this plugin, Manders co-localization was found as an appropriate analysis strategy as this method measures the fraction of co-occurrence of the signal in two channels rather than their correlation [58]. After manual selection of the cell area with the region of interest (ROI) manager, the Manders’ coefficient (MC) was calculated for 28–40 biological replicates for each time point (using three technical replicates). More specifically, MC indicates the fraction of pixels of the αCD44-PEI-MSNPs’ ROI that overlap with the pixels of the LysoTracker ROI, resulting in a value between 0 and 1. One means that 100% of the pixels of the αCD44-PEI-MSNP ROI overlap with the pixels of the LysoTracker channel, with 0 being a 0% pixel overlap.

### 2.13. Doxorubicin Release

The αCD44-PEI-MSNPs were loaded with doxorubicin to a final concentration of 40 μM (αCD44-PEI-MSNP_Dox). A549 cells were incubated with αCD44-PEI-MSNP_Dox for 24 h. After 24 h of incubation, the sample was washed with PBS three times to remove extracellular nanoparticles and kept in HBSS during image acquisition. One sample was immediately measured (with confocal fluorescence microscopy), while two other replicates were placed back in the incubator for the 48- and 72-h time points (after replacing the PBS by cell culture medium). An additional sample was checked after 8 h of nanoparticle incubation, in order to monitor nanoparticle endocytosis. Prior to imaging, the samples were washed three times with PBS and the acidic vesicles were visualized by adding a solution containing LysoTracker^TM^ Deep Red (100-nM final concentration in HBSS) for 15 min. After a final washing step with PBS, the samples were imaged in HBSS using a confocal microscope.

### 2.14. Cytotoxicity Studies

A549 and HepG2 cells were seeded in a 96-well plate at a density of 2 × 10^4^ cells/well. The next day, 25, 50, 100 and 200 nM of Dox and 25, 50, 100 and 200 μg/mL of empty and Dox-loaded αCD44-PEI-MNSPs were added to the A549 cells. Four biological replicates were prepared for each condition. Cells incubated with nanoparticles were washed with PBS after 24 h of nanoparticle incubation to remove the excess of nanoparticles. The PBS was then replaced by fresh medium and the sample was placed in the cell incubator. Then, 72 h after the addition of free Dox or nanoparticles, the cells were washed 3× with PBS and fixed with 4% PFA (in PBS). Cells were incubated with Hoechst 33342 (1 μg/mL) for 1 h. After washing with PBS, the viable cells were imaged using a Lionheart FX automated microscope (BioTek, Santa Clara, CA, USA) implementing a 10× air objective (NA: 0.3) and a high-power LED of 365 nm, combined with a DAPI filter cube. Images were analyzed using the Gen5^TM^ software (version 3.11.19).

### 2.15. Statistical Analysis

The data are displayed as means ± standard deviations and error bars indicate ± standard deviation. A randomization test was used to compare any two groups of values and performed in the online software tool “Plots of Difference” [59]. Statistical significance was reported as * *p* < 0.05, ** *p* < 0.01, and *** *p* < 0.001.

## 3. Results and Discussion

### 3.1. The αCD44-Conjugated Nanoparticles

CD44 is a transmembrane glycoprotein receptor overexpressed in different cancers (e.g., breast, lung, colon, head and neck, and pancreatic cancer [60,61,62,63]) and cancer stem cells [64,65]. Its presence is often associated with high malignancy and chemo-resistance, making it an important cancer biomarker and target for cancer therapy. Hyaluronic acid (HA) is a ligand for the CD44 receptor and has, therefore, been widely exploited as a poly-mer coating in targeted DDSs [2,66,67]. The ligand, HA, is naturally present in the extracellular matrix (ECM), leading to possible competition between HA from the ECM and the HA on the nanoparticles. As an alternative, antibodies with a high affinity for CD44 can be implemented in the DDS to target the receptor [68,69,70].

#### 3.1.1. Synthesis of Azide-Functionalized MSNPs

Our approach to conjugate antibodies to nanoparticles comprised the grafting of different groups on both the antibody and the nanoparticle, separately (Figure 1). As a model for nano-carriers, we used MSNPs. PEI-coated MSNPs were prepared as previously described [49]. Briefly, MSNPs were synthesized using the biphase stratification method and loaded with either Dox or RhoB, for cytotoxicity studies or imaging assays, respectively. The fluorescence spectra of the dye/drug-loaded MSNPs are shown in Appendix A (Appendix A). A PEI layer (Mw = 1.3 kDa) was deposited on the MSNPs (PEI-MSNPs), through electrostatic interactions. In our approach, the PEI layer served two purposes: to provide the nanoparticles with the capability of endosomal escape (as reported by Fortuni et al. [49]) and to functionalize the surface of the NPs with amine groups. These amine groups were used as an anchor for further covalent functionalization. An Azide moiety was covalently linked to the PEI-MSNPs, making use of an NHS ester-PEG_4_-N_3_ linker via NHS ester coupling (N_3_-PEI-MSNPs, Figure 1a). The central poly(ethylene)glycol (PEG) chain provided extra stealth to the DDS, increasing its biocompatibility. The resulting surface Azide groups served as docking sites for the antibodies.

#### 3.1.2. Labelling of CD44 Antibodies

To attach the antibody molecules to the Azide-grafted nanoparticles, the antibodies were functionalized with a DBCO moiety (Figure 1b). Additionally, for visualization purposes, a fluorescent dye (Atto488 or Atto565) was also added, yielding dual-labelled antibodies. As described in the Methods section, labelling of the lysine residues of the antibody was achieved via an NHS ester coupling reaction. The presence of the DBCO moiety was verified via UV-VIS absorption measurements (absorption peak at 309 nm, see Appendix A in Appendix A). To confirm that the addition of DBCO and/or fluorescent dye did not affect the specificity of the antibody, we performed immunostainings of two cell lines, A549 and HepG2 cells, with high and low expression level of CD44, respectively [60]. Before testing the specificity of the DBCO/Atto488-labelled antibodies, the CD44 expression level in both cell lines was evaluated using immunofluorescence. For this, we stained the cells with an unmodified CD44 antibody, followed by a second staining step with a fluorescently labelled secondary antibody. As shown in Figure 2a, CD44 receptor molecules were detected in A549 cells (localized to the cell membrane), while no CD44 immunostaining was visible on the fluorescence images acquired for HepG2 cells. A549 cells stained with the DBCO/Atto488 dual-labelled CD44 antibodies displayed a fluorescence signal on the plasma membrane, indicating that the specificity of the antibody was retained after labelling (Figure 2b). The fluorescence signal present in the nucleus was attributed to the free dye molecules still present in solution after the labelling procedure (Figure 2c).

#### 3.1.3. Synthesis and Characterization of αCD44-Conjugated MSNPs

The antibody was covalently linked on the nanoparticle via copper-free click chemistry between the Azide moiety and the DBCO-labelled antibody (generating αCD44-PEI-MSNPs, Figure 1c). The developed nanoparticles were characterized using scanning electron microscopy (SEM), atomic force microscopy (AFM), zeta potential measurements, and fluorescence microscopy.

SEM images displayed a uniform size and shape homogeneity of the bare MSNPs, PEI-MSNPs and αCD44-PEI-MSNPs (conjugation with αCD44), with a mean diameter of 118, 119 and 124 nm, respectively (Figure 3a). The average size of the bare and PEI-coated MSNPs was in agreement with previous reports [49]. Further, a low degree of aggregation was observed in all the samples. Although there was no significant difference in diameter between the bare MSNPs and the PEI-MSNPs, a significant increase could be observed upon antibody functionalization (*p* < 0.05). These results were confirmed with AFM mea-surements, where an average nanoparticle height of 121, 123 and 128 nm was detected for the MSNPs, PEI-MSNPs and αCD44-PEI-MSNPs, respectively. The corresponding AFM images and plots are displayed in supporting information (Appendix A in Appendix A). The increase in MSNP height upon antibody functionalization indicated a successful attachment to the surface. The small difference can be attributed to the average size of an antibody (in the range of 5 to 15 nm, depending on its orientation).

Zeta potential measurements were used to follow each step of the functionalization process. Bare MSNPs had a negative charge (−37.5 mV) due to the partially deprotonated hydroxyl groups on the MSNP surface. Upon PEI coating, the zeta potential increased to +63 mV as a result of the presence of amine groups in PEI. Subsequent coupling of the Azide moiety was reflected in a decrease in the zeta potential (+31 mV) (Figure 3c). This decrease can be associated with the formation of an amide upon NHS coupling to the amine. Since the amine was positively charged and the resulting amide was neutral, the reaction promoted a decrease in the overall charge. Finally, a further decrease in zeta potential (−6.7 mV) upon antibody grafting could be observed (Figure 3c). The charge of the nanoparticle after antibody conjugation can be related to the isoelectric point (IEP) of the antibody. Since the IEP of IgG antibodies lies between 6.6 and 7.2 [71], antibodies were expected to be negatively charged at neutral pH, which agreed with our zeta potential results and previously reported results [41,72]. To evaluate the uniformity of antibody conjugation, MSNPs were loaded with a dye (RhoB) before functionalization. Since the antibodies were fluorescently labelled (Atto488), the colocalization between the RhoB-loaded MSNPs and the antibody was evaluated with confocal fluorescence microscopy. Figure 3b shows a clear overlap between the Atto488-labelled antibody and the RhoB-loaded MSNPs. This fluorescence data supported the findings obtained from AFM and the zeta potential measurements, indicating the proper conjugation of the antibody.

The colloidal stability of the antibody-conjugated nanoparticles in the presence of serum was investigated using turbidity measurements. Turbidity measurements are a standard technique to detect particle aggregation, and are well described in literature [73]. This technique is based on the scattering of light by a solution of particles, which corresponds to the turbidity of the suspension. The scattered light is detected via the absorbance at 600 nm (OD_600_). This results in a value for the optical density of the samples, which is higher for more turbid samples. CD44 antibody-conjugated nanoparticles, dissolved in milli-Q water, FBS, and DMEM medium with 10% FBS (used for cell culture experiments), were incubated for 24 h at 37 °C. The OD_600_ was determined at time 0 and after 24 h of incubation. The absorbance values are presented in Appendix A (Appendix A) and showed no significant increase in sample turbidity after 24 h of incubation in all the tested solutions, indicating that the particles were not prone to aggregation in the presence of serum proteins.

### 3.2. Selectivity and Efficiency of αCD44-Functionalized Particles

#### 3.2.1. Targeting Capability of αCD44-PEI-MSNPs

To assess the targeting capability of the antibody-conjugated MSNPs towards CD44-overexpressing cells, FITC-encapsulated MSNPs were used to monitor the cellular uptake via fluorescence microscopy. In this case, targeting capability refers to the selective uptake of the CD44 antibody-conjugated nanoparticles to CD44 receptor-overexpressing cells. The targeting efficiency was tested by comparing the number of nanoparticles internalized in A549 cells (human lung carcinoma cells, with CD44 receptor overexpression [74]) and HepG2 cells (liver carcinoma, with low CD44 receptor expression [75]). Both cell lines were incubated for 6 h with MSNPs presenting different functionalization: no coating (MSNPs), only a PEI coating (PEI-MSNP) and both the PEI and antibody functionalization (αCD44-PEI-MSNPs). As a control, the cells were also incubated with a nanoparticle conjugated to a non-specific IgG antibody (IgG-PEI-MSNP). Afterwards, the medium was refreshed to avoid further nanoparticle internalization and the cells were incubated overnight (24 h incubation in total). The internalization was quantified by confocal fluorescence imaging after staining the plasma membrane with a lipid intercalating dye (DiR). The fluorescence images are shown in Figure 4. The uptake of bare MSNPs was minimal in both cell lines (Figure 4a,e), while a coating with PEI resulted in an increase in internalization in both A549 and HepG2 (Figure 4b,f). The increase in cellular uptake of nanoparticles upon PEI functionalization was in agreement with previous reports and can be attributed to the positive charge of PEI-MSNPs, leading to a higher interaction with the negatively charged plasma membrane, which facilitates nanoparticle internalization [49,51,76,77]. It is important to note that this enhanced uptake of PEI-coated MSNPs was cell line unspecific. Thanks to the charge drop (from +63 mV to −6.7 mV), this unspecific internalization was minimized upon antibody conjugation to MSNPs. As such, uptake of αCD44-PEI-MSNPs in HepG2 was lower, similar to the internalization of bare MSNP (Figure 4g). On the other hand, A549 cells incubated with αCD44-PEI-MSNPs displayed a high number of internalized particles, especially when compared to bare MSNP (Figure 4e). A quantitative mean fluorescence intensity of the internalized nanoparticles revealed that there were 20 times more particles inside A549 cells when compared to HepG2 cells (Figure 4i). Incubation with the non-specific IgG-conjugated nanoparticles resulted in an uptake comparable to bare MSNPs in both A549 and HepG2 cells. This low internalization could be explained by the non-specificity of this antibody to membrane proteins in these cell lines and the negative zeta potential (−11.8 mV) of the resulting IgG-conjugated nanoparticles (which minimized their non-specific uptake). To assure that the observed increase in cellular uptake in CD44-overexpressing cells was not linked to the specific cell lines used, we checked the uptake of αCD44-PEI-MSNPs in two fibroblast cell lines, NIH3T3 (mouse embryonic fibroblasts with a low CD44 expression [78,79]) and BJ1-hTERT (human fibroblasts with high CD44 expression [80,81]), using the same experimental approach (Appendix A). The respective CD44 expression in the two cell lines was validated with a standard immunofluorescence staining of CD44 (Appendix A, Appendix A). The discrepancy between the uptake of αCD44-PEI-MSNPs in fibroblasts with different expression levels of CD44 confirmed the targeting efficiency of our nanoparticles after conjugation with the CD44 antibody.

#### 3.2.2. Intracellular Trafficking

One of the main bottlenecks faced by DDSs is their entrapment in acidic vesicles and subsequent degradation, significantly limiting the overall efficiency of the DDS [82,83,84]. To this end, strategies have been developed to facilitate an endosomal escape, releasing the nanoparticles (or their cargo) into the cytoplasm. We previously showed that the addition of a PEI shell leads to the release of the nanoparticles in the cytoplasm [49]. To verify that conjugation to an antibody does not affect the endosomal escape capability of the PEI layer, the co-localization of αCD44-PEI-MSNPs with lysosomes was monitored using fluorescence microscopy. Briefly, A549 cells were incubated with RhoB-loaded αCD44-PEI-MSNPs for 24 h, after which the medium of the samples was refreshed, in order to avoid further nanoparticle internalization (i.e., only nanoparticles that were endocytosed within the first 24 h of incubation were followed). At each time point (24, 48 and 72 h after particle addition), the acidic vesicles were stained with LysoTracker Deep Red and the samples were imaged. Figure 5 shows representative fluorescence images at the different time points. To quantify the co-localization between the acidic vesicles and the αCD44-PEI-MSNPs through time, the Manders’ co-localization coefficient was calculated (Appendix A, Appendix A). Briefly, an intensity-based threshold was used to calculate the areas of the image corresponding to nanoparticles and to lysosomes. The Manders’ coefficient (MC) calculates the degree of overlap between objects in different channels (with 0 being no overlap and 1 indicating a complete overlap). Fluorescence images showed that, within 24 h, almost all nanoparticles were located inside the acidic vesicles (Figure 5a). While at 24 and 48 h, no relevant escape was observed with MC of 0.88 and 0.82 (Appendix A), a significant decrease in the co-localization of nanoparticles and lysosomes was found after 72 h (MC of 0.66), indicating that after 72 h a considerable number of αCD44-PEI-MSNPs were localized outside the lysosomes (Figure 5c). Overall, these results suggest that the presence of αCD44 on the surface of the nanoparticles does not affect the endosomolytic activity of the PEI coating as most of nanoparticles were able to dissociate from the lysosomes within 72 h. This was validated by the significant difference in MC found between 48 and 72 h (Appendix A). The endosomal escape rate was, however, slower than MSNPs coated with only PEI, where the majority of the particles escaped the lysosomes within 48 h (previously reported by our group [49]). Since the proton sponge effect is linked to the amine groups of the PEI, this delay might be associated with the reduced number of amine groups available, as part of these are used for N3 functionalization and further coupling to the antibodies.

#### 3.2.3. Intracellular Release of Doxorubicin

As shown in the previous section, a considerable number of the internalized αCD44-PEI-MSNPs were able to escape the acidic vesicles. Aside from the ability to escape the lysosomes, a higher efficiency requires that enough cargo can be released into the cytoplasm in a controlled fashion. To monitor the drug release after cellular uptake, doxorubicin was used as drug model and encapsulated in the pores of MSNPs prior to coating (αCD44-PEI-MSNPs_Dox). Dox is a cytostatic anticancer drug that is used to treat different types of cancer, for instance, leukemia, lymphoma, and breast cancer. Its mechanism of action is based on the intercalation with the DNA, resulting in cell death [85]. Due to the fluorescent nature of Dox, its intracellular localization could be monitored over time with confocal fluorescence microscopy. To show that αCD44-PEI-MSNPs_Dox carried Dox into the target cell and exhibited a controlled intracellular drug release, A549 cells were incubated with αCD44-PEI-MSNPs_Dox for 8, 24, 48 and 72 h. To avoid further nanoparticle internalization, the medium of the samples was refreshed 24 h after the addition of the nanoparticles. The acidic compartments were stained with LysoTracker Deep Red (magenta, Figure 6) to analyze the cargo (Dox, cyan) release/leakage from the lysosomes. While after 8 and 24 h, most of the Dox was still retained inside lysosomes (indicated by the high overlap between the Dox and the lysosomes, Figure 6a,b), increasing amounts of Dox discharged into the cytosol were observed after 48 and 72 h (Figure 6c,d). The overlap between the acidic vesicles and Dox molecules at 8 and 24 h indicated that, at this time point, Dox was presumably still inside the pores of the MSNPs or being slowly released inside the acidic vesicles. At an acidic pH, the silica hydroxyl groups are protonated, hindering the electrostatic interaction between the PEI and the silica surface. Therefore, we hypothesized that the decreased pH inside the lysosomes aids in the release of the shell and facilitates the consequent Dox release from the mesoporous silica pores. Accordingly, increasing amounts of Dox were detected in the cytoplasm over 48 to 72 h (Figure 6c,d). Moreover, as shown in Figure 6d, the cells’ morphology suggested cellular death, indicating that the cells were being killed by the successful release of the drug.

#### 3.2.4. Cytotoxicity Studies

To evaluate the cytotoxicity of the antibody-coated MSNPs, the viability of A549 and HepG2 cells was investigated after 72 h of incubation with pure Dox, Dox-loaded nanoparticles (αCD44-PEI-MSNPs@Dox), and empty nanoparticles (αCD44-PEI-MSNPs). After 24 h of incubation with the nanoparticle/drug solution, the medium was refreshed to avoid the continued uptake of nanoparticles/drug in the medium and to better mimic the physiological scenario of nanoparticle clearing from the bloodstream. Different concentrations of Dox and nanoparticles were added, ranging from 25 nM to 200 nM for free Dox and 25 to 100 μg/mL for the nanoparticles (Figure 7). For both cell lines, empty αCD44-PEI-MSNPs could be considered non-toxic, reaching a minimum of 92% cell viability at the highest concentration (100 µg/mL). For both cell lines, a dose-dependent response on the cell viability was observed for free Dox. We found an evident difference in the cell-killing effect between the free Dox and the Dox-loaded nanoparticles (*p* < 0.05) for A549 cells. Dox-loaded αCD44-PEI-MSNPs exhibited dose-dependent cytotoxic effects for A549 (reaching a minimum of 38% of viability at the highest concentration). This enhanced cytotoxicity of the Dox-loaded nanoparticles is linked to the high internalization rate of the αCD44-PEI-MSNPs followed by endosomal release and specific drug release in the cytoplasm. Importantly, the negligible internalization of αCD44-PEI-MSNPs in HepG2 cells resulted in the absence of toxic effects when using Dox-loaded particles (viability similar to the control). The high cell viability also indicates that the drug did not leak from the nanoparticles into the extracellular environment.

From these results, it can be concluded that Dox-loaded αCD44-PEI-MSNPs exhibited high toxicity to CD44-overexpressing cells only (higher than free Dox), while only limited to no toxicity was observed for the empty carrier. Consequently, a satisfying therapeutic efficiency can be concluded, through an efficient nanoparticle internalization and high drug payload delivery targeted to the cancer cell cytosol.

### 3.3. The αEGFR-Conjugated Nanoparticles

To prove the versatility of our antibody coating strategy, we targeted next a second cancer marker, EGFR. EGFR overexpression is related to different types of cancers and is often associated to a poor prognosis for the patient. Especially in glioblastoma, lung, and breast cancers, where EGFR stimulates tumor growth [44]. To this end, EGFR has emerged as a popular target for cancer cell-specific therapies. As a result, many nanoparticles targeting EGFR were developed, either via nanoparticle conjugation with its ligand, EGF, or by attaching EGFR antibodies [86,87,88,89].

#### 3.3.1. Labelling of the EGFR Antibodies

For labelling the EGFR antibodies, the same method was used as for the CD44 antibodies (Section 3.1.2). Similarly, labelling with DBCO was performed, enabling the conjugation of DBCO-EGFR antibodies to the Azide-coated nanoparticles. For fluorescence microscopy purposes, the EGFR antibodies were also conjugated to a dye (Atto565). The dual labelling of the EGFR antibodies with DBCO and Atto565 was carried out via an NHS ester coupling reaction with the antibodies’ lysine residues (as described in Methods). We checked whether the specificity of the EGFR antibody was retained after dual labelling (with Atto565 and DBCO) by performing immunofluorescence on two cell lines, A431 and Hek293 cells, presenting high and low expression of EGFR, respectively [90,91]. First, the EGFR expression in the chosen cell lines, A431 and Hek293, was determined via immunofluorescence with the unmodified EGFR antibody (followed by staining with a fluorescently labelled secondary antibody, goat-anti-mouse IgG AF488). As shown in Figure 8a, EGFR was visualized on the plasma membrane of A431 cells, while no EGFR could be detected in Hek293 cells. A431 cells stained with the DBCO/Atto565-labelled EGFR antibody also displayed the plasma membrane-localized signal, proving that the specificity of the EGFR antibody was not hindered by our labelling protocol (Figure 8b). A nuclear background could be observed, associated to free Atto565 molecules, similar to what was observed for the Atto488 molecules (see Section 3.1.2).

#### 3.3.2. Synthesis and Characterization of αEGFR-Conjugated MSNPs

The EGFR antibody was attached to the Azide-coated nanoparticles via the same approach (αEGFR-PEI-MSNPs). Since the EGFR antibody was labelled with an Atto565 dye (as described in the previous section), to avoid spectral overlap during fluorescence microscopy studies, Azide-conjugated nanoparticles were labelled with Fluorescein (FITC). FITC was covalently attached to the silica matrix, as stated in the Methods section. The resulting αEGFR-conjugated nanoparticles were characterized via SEM, AFM, zeta potential measurements, and fluorescence microscopy.

Similar to αCD44-conjugated particles, αEGFR-PEI-MSNPs displayed a uniform size and shape homogeneity (Figure 9a). With a mean diameter of 122 nm (as derived from the SEM data), αEGFR-PEI-MSNPs were significantly bigger than the PEI-MSNPs (Figure 3a) (*p* < 0.05) because of the EGFR layer. AFM measurements revealed an average nanoparticle diameter of 127 nm, which agreed with the SEM results (Appendix A). Zeta potential measurements were performed to check the different steps of nanoparticle functionalization. The zeta potential data of the bare MSNPs, PEI-grafted MSNPs (PEI-MSNPs), and Azide-conjugated MSNPs (N_3_-PEI-MSNPs) were already discussed in Section 3.1.3. (Figure 3c). The αEGFR-PEI-MSNPs displayed a negative zeta potential (−11.6 mV), similar to the one of the MNSPs after conjugation with the CD44 antibody (αCD44-PEI-MSNPs, −6.7 mV) (Figure 9b). Additionally, in order to confirm uniform binding of the EGFR antibody, MSNPs were encapsulated with a fluorescent dye (FITC), while the EGFR antibody was labelled with another dye (Atto565). Accordingly, the co-localization between the EGFR antibody and the MSNPs could be determined via confocal fluorescence microscopy measurements. These data confirmed proper attachment of the EGFR antibody and uniform coverage of the nanoparticles with antibody molecules (Figure 9c).

#### 3.3.3. Targeting Capability of αEGFR-PEI-MSNPs

The uptake of nanoparticles with different coatings (MSNPs, PEI-MSNPs, αEGFR-PEI-MSNPs, and IgG-PEI-MSNP) was compared between A431 (epidermoid carcinoma with high EGFR expression [86]) and Hek293 cells (human embryonic kidney cells with low EGFR expression [91]). In agreement with the results of the preceding experiment, bare MSNPs were internalized in A431 and Hek293 in a low amount (Figure 10a,e), and a similar increase in uptake could be detected for MSNPs with a PEI layer (Figure 10b,f). As predicted, conjugation of the EGFR antibody resulted in a significant discrepancy in particle uptake between A431 and Hek293 cells (Figure 10c,g,i). Hek293 displayed a similar uptake behavior for the αEGFR-PEI-MSNPs and the bare MSNPs (Figure 10g) while, in A431 cells, there was an obvious increase in the number of αEGFR-PEI-MSNPs internalized, especially when compared to bare MSNPs (Figure 10c). This increase can be attributed to the specific recognition of the EGFR receptor by the αEGFR-PEI-MSNPs, proving cell specificity of these nanoparticles. Similar to A549 and HepG2, the incubation of A431 and Hek239 cells with the non-specific IgG-conjugated nanoparticles resulted in an internalization comparable to bare nanoparticles. As mentioned in Section 3.2.2, this is a consequence of the non-specificity of the IgG towards membrane proteins expressed in these cell lines and the negative zeta potential (−11.8 mV) of the resulting IgG-conjugated nanoparticle (minimum non-specific uptake). To further evaluate the specificity of the antibody-conjugated nanoparticles, expression of the EGFR receptor in Hek293 was induced by transiently transfecting Hek293 cells with a plasmid encoding an EGFR-HaloTag^®^ fusion construct (see Methods section for details). A fluorescent ligand (Alexa Fluor 488 HaloTag^®^ ligand) was used to identify the EGFR-expressing Hek293 cells via confocal fluorescence microscopy (Appendix A in Appendix A). Quantitative analysis of the fluorescence intensity revealed that Hek239 cells expressing EGFR molecules exhibited a higher uptake of αEGFR-PEI-MSNPs (Figure 10i). This demonstrates that the internalization of αEGFR-PEI-MSNPs is linked to the presence of the EGFR receptor on the cell membrane, further confirming the selectivity of the developed nanoparticles.

## 4. Conclusions

In this work, we showed a facile method for the conjugation of different antibodies to nanoparticles using copper-free click chemistry. Here, mesoporous silica nanoparticles (MSNPs) were chosen as drug carriers and the base for further functionalization; however, other nanoparticles can be used. Similarly, the Doxorubicin loaded inside the MSNPs can be substituted by other drugs for different therapeutic applications.

The first step in our approach was to coat the nanoparticles with a PEI layer. The PEI amine groups provide the anchor for the covalent attachment of an Azide moiety, to which the DBCO-labelled antibodies were covalently linked via a simple click reaction. Importantly, click chemistry can be carried out under physiological conditions, without using any catalyst. Furthermore, the presence of the PEI layer reduced the effect of nanoparticle entrapment in the acidic vesicles (by taking advantage of the proton sponge effect) and supported the controlled drug release into the cancer cell.

All existing antibodies can be labeled with a DBCO group in the same controllable way (via an NHS-ester coupling with their amine groups) and their conjugation to the nanoparticles only requires the presence of an amine group on the nanoparticle surface. Therefore, this approach to produce antibody-conjugated nanoparticles is extremely versatile and has the potential to be widely applied. The antibody that is conjugated can be easily adapted (based on the expressed cancer markers) and the type of nanocarrier can be changed (as long as surface amine groups are present). Here, the versatility of our coating strategy was demonstrated with two different antibodies, a CD44 and an EGFR antibody, both showing excellent selectivity towards CD44- and EGFR-overexpressing cells, respectively. This simple method can significantly contribute to the field of personalized cancer therapy, where the treatment should be customized according to the cancer markers present in the tumor. In this regard, a variety of antibodies can be easily “clicked-on” for efficient targeting purposes.

## Figures and Tables

**Figure 1 pharmaceutics-13-02153-f001:**
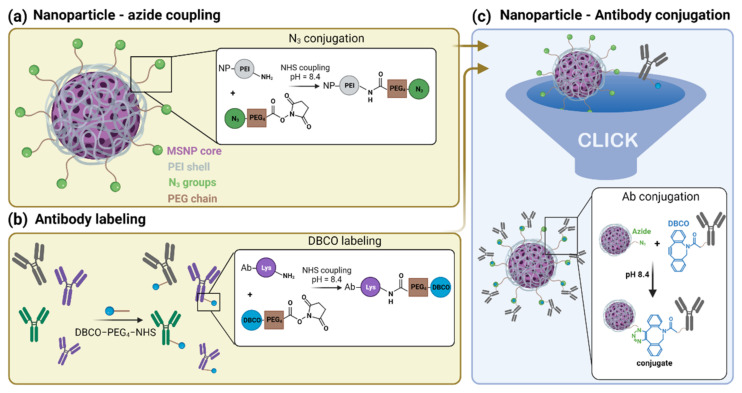
Schematic representation of the antibody-conjugated nanoparticle preparation: (**a**) Mesoporous silica nanoparticles (MSNPs) loaded with the drug (Doxorubicin) and coated with a polyethyleneimine (PEI) layer (gray) are grafted with an Azide moiety (N_3_, green) using NHS ester coupling. (**b**) The lysine residues (purple) of different antibodies were labelled with a DBCO moiety (blue) via NHS ester coupling. (**c**) Click chemistry reaction of the Azide-functionalized nanoparticle with the DBCO-labelled antibodies resulting in the final antibody-conjugated nanoparticle.

**Figure 2 pharmaceutics-13-02153-f002:**
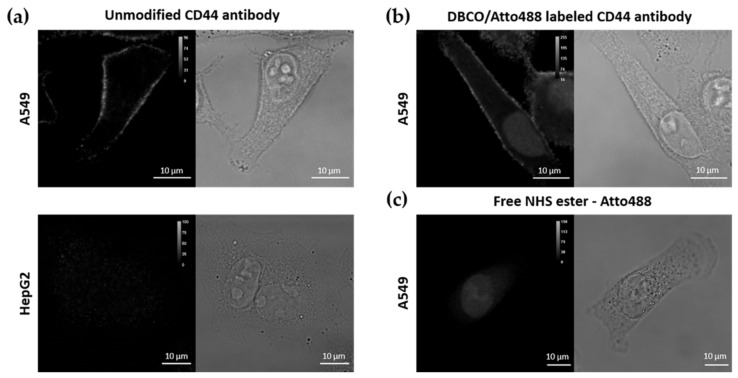
Immunofluorescence (IF) using (**a**) the unmodified primary rat anti-human CD44 antibody and a secondary donkey anti-rat IgG Alexa Fluor 488 antibody on A549 and HepG2 cells, (**b**) dual DBCO/Atto488-labelled CD44 antibodies in A549 cells and (**c**) free NHS ester-Atto488 molecules. Scale bar is 10 µm.

**Figure 3 pharmaceutics-13-02153-f003:**
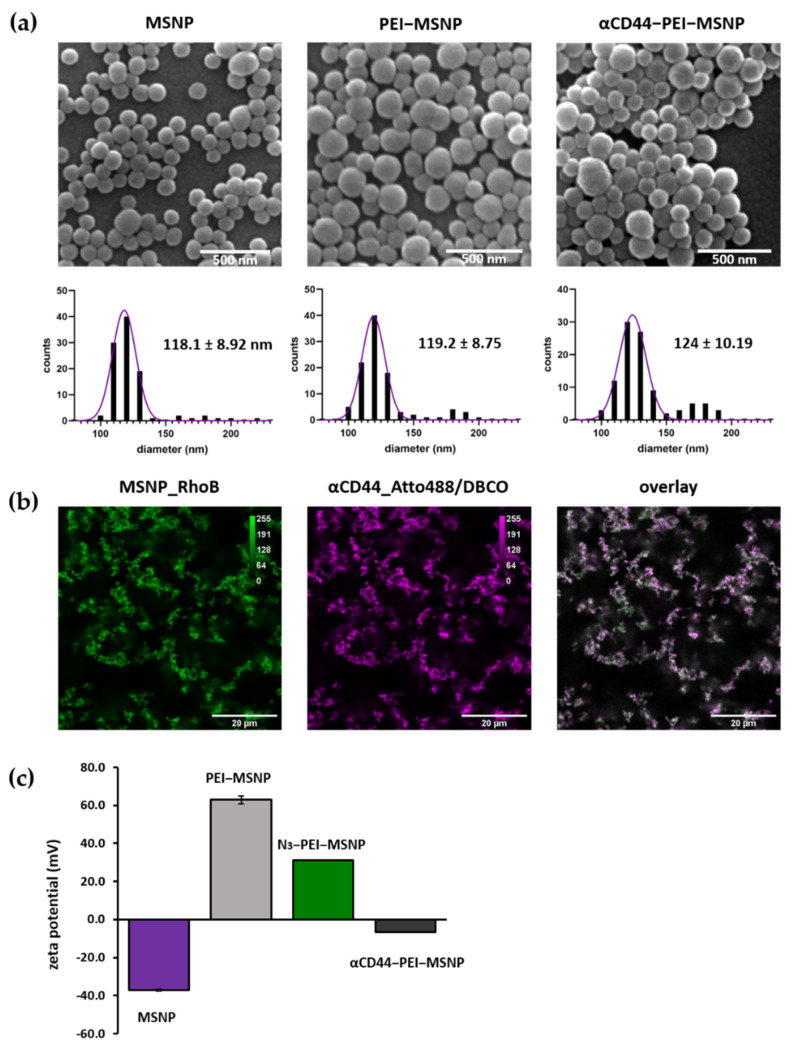
Characterization of bare (MSNPs), PEI-coated (PEI-MSNPs), Azide-functionalized (N_3_-PEI-MSNPs), and CD44-conjugated (αCD44-PEI-MSNPs) mesoporous silica nanoparticles. (**a**) Representative SEM images and size distribution of the imaged particles. Values shown as mean ± SD. Scale bar is 500 nm. (**b**) Confocal fluorescence images of αCD44-PEI-MSNPs in which MSNPs were loaded with RhoB (first panel, green), while an Atto488 (and DBCO) label was conjugated to the CD44 antibody (second panel, magenta). An overlay is displayed in the third panel. Scale bar is 20 µm. (**c**) Zeta-potential measurements given as mean ± SD.

**Figure 4 pharmaceutics-13-02153-f004:**
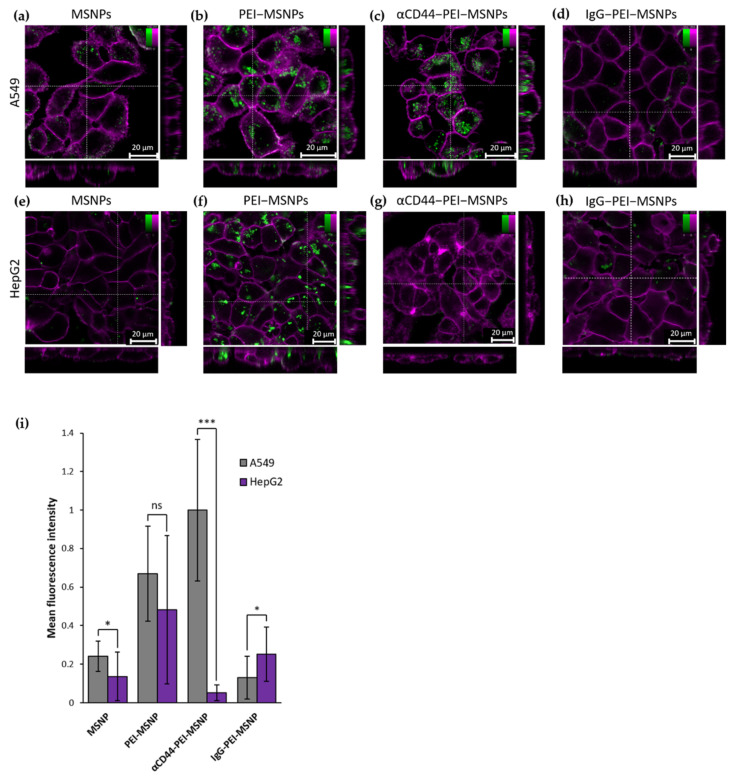
Confocal fluorescence microscopy images showing the influence of different MSNP coatings on the uptake of nanoparticles in A549 and HepG2 cells. Internalization of bare MSNP (**a**,**e**), PEI-coated MSNPs (PEI-MSNP, panels (**b**,**f**)), CD44-functionalized MSNPs (αCD44-PEI-MSNP, panels (**c**,**g**)), and IgG-functionalized MSNPs (IgG-PEI-MSNP, panels d,h) in A549 cells (**a**–**d**) and HepG2 cells (**e**–**h**). Nanoparticles were loaded with Fluorescein (FITC, green) and the plasma membrane was stained with DiR (magenta). The central square represents a single xy plane, while the bottom and left panels are the xz and yz cross-sections, indicated by the dashed lines. Scale bar is 20 μM; color bars display the intensity values. (**i**) Normalized mean fluorescence intensity of MSNP, PEI-MSNP, αCD44-PEI-MSNPs, and IgG-PEI-MSNPs internalized in A549 and HepG2 (20 cells per condition). The data were analyzed using Fiji (see Methods section for details). With ns: not significant, * (*p* < 0.05), *** (*p* < 0.001).

**Figure 5 pharmaceutics-13-02153-f005:**
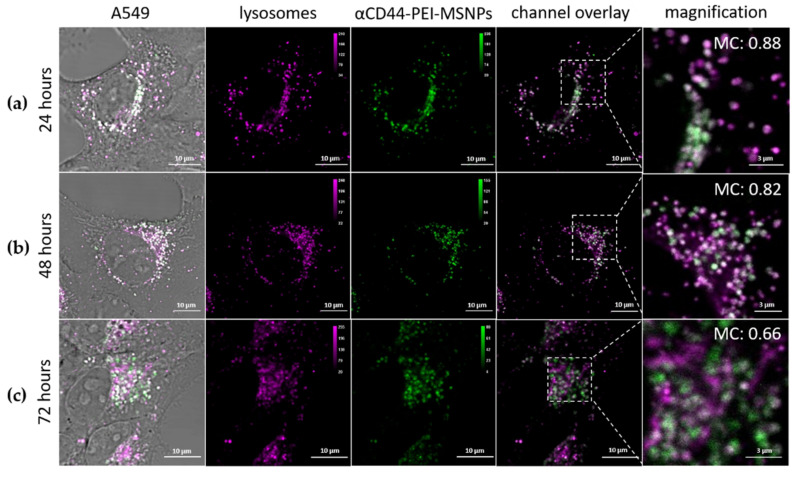
Intracellular localization of RhoB-loaded αCD44-PEI-MSNPs (green) with respect to the lysosomes (Lysotracker Deep Red, magenta) over time. A549 cells were incubated with αCD44-PEI-MSNPs (final concentration of 50 μg/mL) for (**a**) 24 h (**b**) 48 h and (**c**) 72 h. The first column shows a complete merge including the transmission image, also displaying the cell areas. In the second and third columns, the lysosomes and nanoparticles are depicted, in purple and green, respectively. In the fourth and fifth columns, a channel overlay and respective magnification are shown. Scale bar is 10 μm in the main images and 3 μm in the magnified images.

**Figure 6 pharmaceutics-13-02153-f006:**
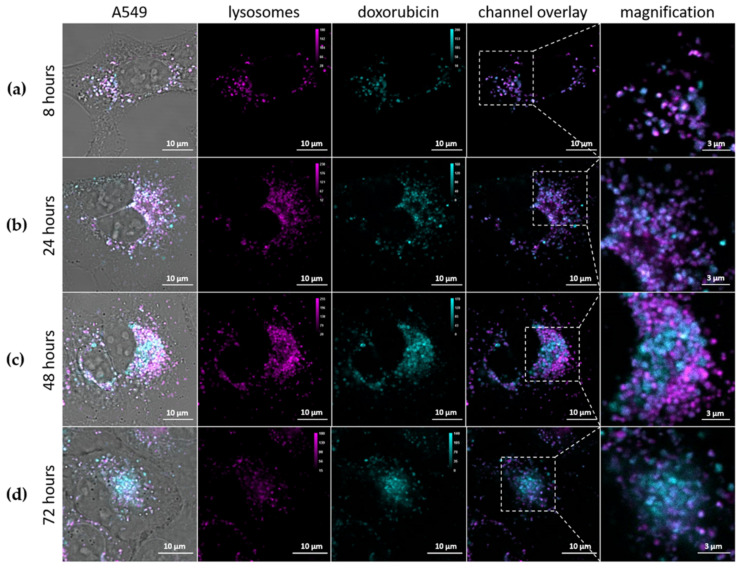
Intracellular release of Dox (cyan) from αCD44-PEI-MSNPs_Dox with respect to the lysosomes (magenta) over time. A549 cells were incubated with αCD44-PEI-MSNPs_Dox (final concentration of 50 μg/mL) for (**a**) 8 h, (**b**) 24 h, (**c**) 48 h and (**d**) 72 h. The first column shows a complete merge including the transmission image, also displaying the cell contours. In the second and third columns, the lysosomes and Dox are depicted, in pink and cyan, respectively. In the fourth and fifth columns, a channel overlay and respective magnification are shown. Scale bar is 10 μm in the main images and 3 μm in the magnified images.

**Figure 7 pharmaceutics-13-02153-f007:**
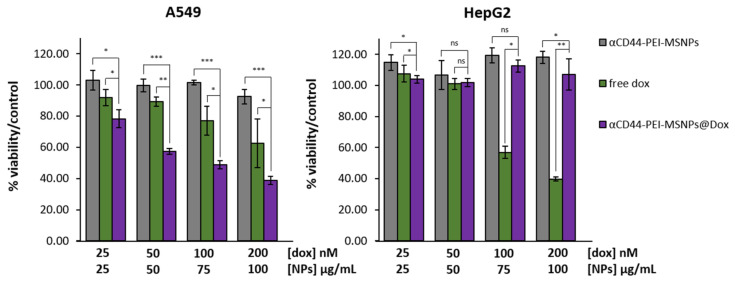
Viability of A549 and HepG2 cells after 72-h incubation with different concentrations of free Dox, Dox-loaded αCD44-PEI-MSNPs, and empty αCD44-PEI-MSNPs. A549 and HepG2 cells that were not incubated with particles or drug were used as a control and represent 100% viability (data not shown). Dox concentrations are expressed in nM while nanoparticle concentrations are in µg/mL. Estimated Dox concentration loaded in the nanoparticles was 50 μM. Error bars indicate ± SD, with ns meaning not significant; * (*p* < 0.05), ** (*p* < 0.01), and *** (*p* < 0.001), n = 4.

**Figure 8 pharmaceutics-13-02153-f008:**
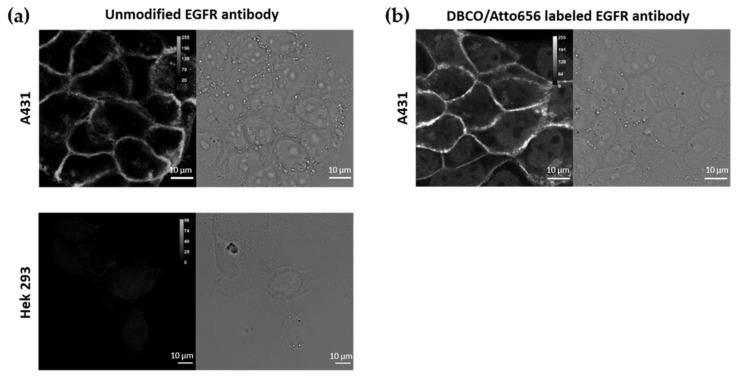
Immunofluorescence (IF) using (**a**) the unmodified primary mouse anti-human EGFR antibody followed by a labelled secondary antibody, goat anti-mouse IgG AF488, on A431 and Hek293 cells; (**b**) DBCO/Atto565-labelled EGFR antibodies in A431 cells. Scale bar is 10 µm.

**Figure 9 pharmaceutics-13-02153-f009:**
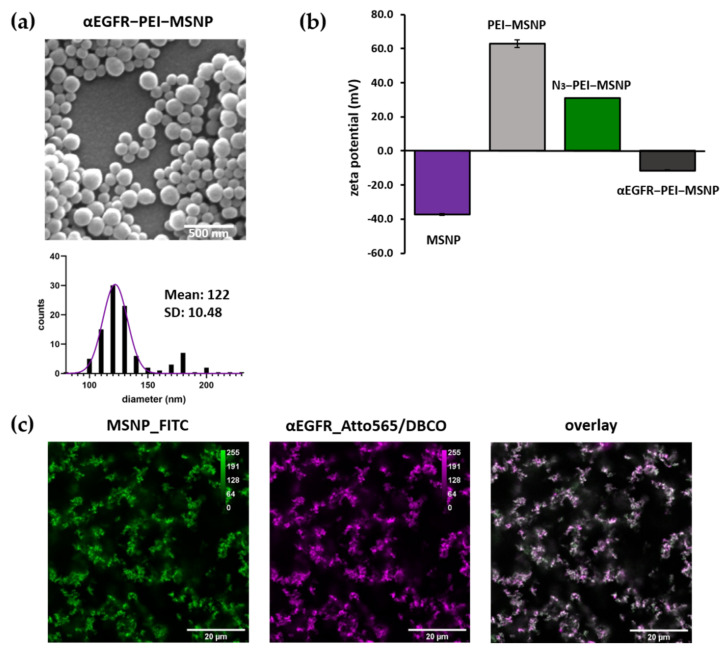
Characterization of EGFR-conjugated (αEGFR-PEI-MSNPs) mesoporous silica nanoparticles. (**a**) Representative SEM image and size distribution of the imaged particles (values shown as mean ± SD). Scale bar is 500 nm. (**b**) Zeta-potential measurements. (**c**) Confocal fluorescence images of sedimented nanoparticles on glass showing the overlay (third panel) of FITC-encapsulated MSNPs (first panel, green) and the Atto565/DBCO-labelled EGFR antibody (second panel, magenta).

**Figure 10 pharmaceutics-13-02153-f010:**
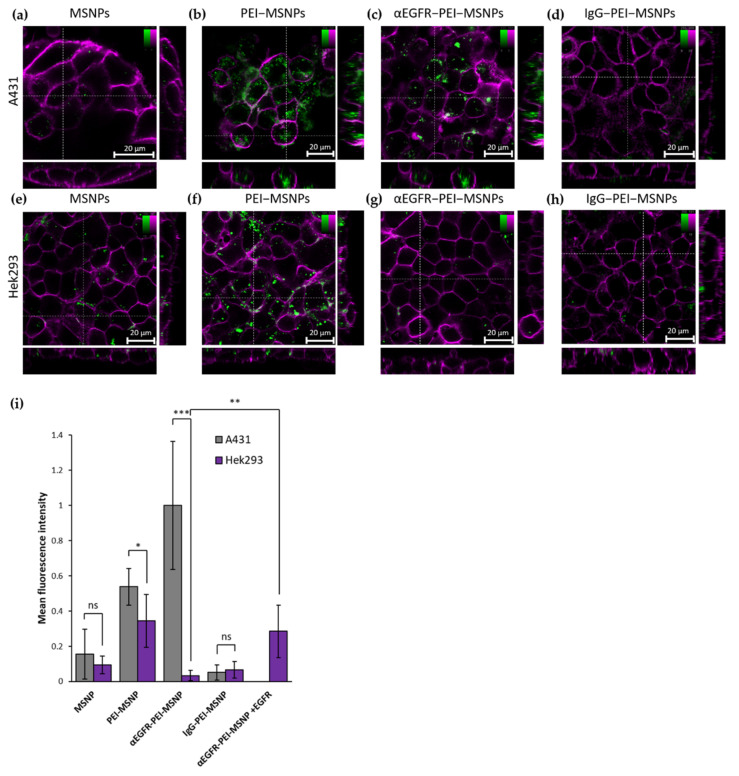
Confocal fluorescence microscopy stacks showing the influence of different MSNP coatings in A431 and Hek293 cells. (**a**–**d**) Internalization of MSNP, PEI-MSNP, αEGFR-PEI-MSNP, and IgG-PEI-MSNP in A431 cells (from left to right). (**e**–**h**) Internalization of MSNP, PEI-MSNP, αEGFR-PEI-MSNP, and IgG-PEI-MSNP in Hek293 cells (from left to right). Nanoparticles (encapsulated with fluorescein) are displayed in green and the plasma membrane is stained with DiR (magenta). Scale bar is 20 μM; color bars display the intensity values. (**i**) Normalized mean fluorescence intensity of MSNP, PEI-MSNP, αEGFR-PEI-MSNPs, and IgG-PEI-MSNPs internalized in A431 and Hek293 cells and αEGFR-PEI-MSNPs in Hek293 cells transiently expressing EGFR (Hek293 + EGFR, 20 cells per condition). The data were analyzed using Fiji (see Methods section for details). With ns: not significant, * (*p* < 0.05), ** (*p* < 0.01), and *** (*p* < 0.001).

## Data Availability

The data presented in this study are available on request from the corresponding authors.

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
