# Peer review of "Versatile and Robust Method for Antibody Conjugation to Nanoparticles with High Targeting Efficiency"

_pharmaceutics, 2021, doi:10.3390/pharmaceutics13122153_

Round 1

Reviewer 1 Report

Dear Authors.

It was my pleasure to read a paper titled “Versatile and Robust method for Antibody Conjugation to 2 Nanoparticles with High Targeting Efficiency”. This paper describes a conjugation strategy of antibodies to nanoparticles using click chemistry. Authors characterized the obtained nanoparticles and demonstrated that conjugation of CD44 and EGFR antibodies leaded to decrease of Z-potential with no size changes. Further, author characterized internalization of nanoparticles either bare, PEI and PEI-antibody coated and demonstrated that, according to the previous author’s data, PEI induces internalization whereas antibody conjugation represses internalization for CD44 or EGRF negative cell line Hepg2 whereas in the EGFR positive A431 and CD44 positive A549 cells Np’s were internalized and in NIH-3T3 CD44 decorated NPs colocalized with membrane. The last part of the study was demonstration of endosomal escape in the CD44-PEI-MSNP.

I would ask for two experiments. First – to better demonstrate MSNP-Ab specificity, I would include at least the first experiment and either one of the second and third:

  1. Nonspecific IgG control conjugated to MSNP, - probably the easiest thing to do.
  2. Target (CD44 or EGFR) overexpression in non-expressing cells following by the same experiment as you already did – just purchase a plasmid from any company such as Addgene.
  3. The gold standard – but, I guess, could be too difficult: knockdown/knockout of the targets.

Second – if it is not already known, is it possible to compare endosomal escape of PEI coated and Ab coated MSNPs as it is shown in Figure S6.

Thank you.

Good luck!

Author Response

The reviewer can find the point-by-point response in the attachment. 

Reviewer 2 Report

The authors have a robust, general and interesting method for the conjugation of nanoparticles to antibodies using well established chemistry that is used in the conjugation of oligonucleotide and drug-linkers as part of the ADCs. 

They have shown that this design is of interest trying to address both the lysosomal escape of agents and the specificity of the nanoparticles.  The article is well written and has a strong foundation of data to support their claims.     The fluorescence microscopy work is thorough and shows conjugation.   

  There was a lot of work showing specific internalization with positive and negative cell lines but the authors in section 3.2.4 discussing viability do not include a non-expressing cell lines.  This is especially important, the statement line 521 claims therapeutic efficiency but this would involve a difference between positive and a control.  If at the concentration given the negative cell lines shows activity than the therapeutic efficiency is not high but if it does than this increases the power of the paper significantly.  As selectivity is the goal of this nanoparticular conjugate not showing it in a viability assay is interesting and no comments offered about it.   Showing this is critical to show selectivity but even if selectivity is not as expected it does not change this DBCO-Click nanoparticles is a rapid conjugation method and the nanoparticles could be modified differently with any azide containing group to allow similar facile conjugation to mAbs.

Other points:

Line 62 ADCs are growing extensively used is a strong statement as currently they have very specific label and would tone down the statement.  I would also include a recent review article on ADC as reference 17-19  are related to specific products and not a where the field is at the time of writing. 

Line 130 to 139:  There is no mention of how DBCO/mAb is present this data could be useful and can be obtained by LC-MS and this number should be included.   

Line 214 used an anti-rat IgG it wasn't clear on line 113 and 114 what species the mAb are? Please include species in description (rat-monoclonal, etc) to help the reader.  For EGFR (mouse monoclonal) this was presumably done with a different secondary using same procedure line 550-551?  If so specifying this would help the reader.

Line 356 to 358:  This statement is incorrect, organic azide are neutral species (zwitterionic, contains a Positive and Negative charge) when attached to carbon .  The decrease of Zeta potential is due to to the amine(+) at that pH being functionalized with the NHS ester to an amide (neutral).  Please revise statement.  Could this be used to determine amount of average functionalization of nanoparticle by the organic azide?  Along the same line no zeta potential was given for the mAb and if known could this also be used to estimate average functionalization of the mAb to nanoparticle?  

Figure 5 and 6, concentration is found in experimental but would be useful to include in figure caption and text for quick reference of how much nanoparticles are there.  

   Section 3.2.4:  It is stated the nanoparticles are flushed after 24h to remove nanoparticles but there is no reference of explanation of why this is important for a reader not an expert in nanoparticle delivery.   Is this because it clears rapidly and mimics real world conditions, or other practical reasons?

Generally this is a thorough paper that has done work on improving delivery of nanoparticles thinking about specificity, trafficking and need to be able to apply to multiple targets and tune properties rapidly.  The reviewer recognizes the work involve in generating all this data and this work will be of interest to the broad readership of this paper. 

Author Response

The reviewer can find a point-by-point response in attachment.

Reviewer 3 Report

Zundert et al. have demonstrated a metal free click chemistry approach to decorate nanoparticles with antibodies. While the problem authors are trying to tackle continues to be addressed by various means, this approach is innovative and introduces versatility in the system. Authors have reported interesting findings, however, some concerns need to be addressed for the manuscript to be deemed suitable for publication.

  1. As majority of the manuscript focuses on targeting receptors associated with solid tumors, author should highlight some unique challenges associated with solid tumors- dense ECM, high interstitial fluid pressure etc. in the introduction section. Following literature reviews would serve as a good starting point:
    • Martin, John D., et al. "Improving cancer immunotherapy using nanomedicines: progress, opportunities and challenges." Nature Reviews Clinical Oncology 17.4 (2020): 251-266.
    • Ukidve, A., Cu, K., Kumbhojkar, N., Lahann, J. and Mitragotri, S., 2021. Overcoming biological barriers to improve solid tumor immunotherapy. Drug Delivery and Translational Research, pp.1-26.
  2. Authors should report and compare mean fluorescent intensity for Fig 2 (a,b,c). As of now, the purpose described in lines 315-317  is not completely justified.
  3. Lines 350-351 points to a potential variability in conjugation of the antibody. Have the authors performed specificity assays , such as ELISA, to demonstrate that activity/ CDR region of the antibody is not affected by conjugation?
  4. Figures 4,5,6,8 and 10 provide a very qualitative insight into the activity of these conjugated nanoparticles. Mean fluorescent intensity for all the confocal images should be calculated and included as part of the these figures to substantiate the claims made in these sections.
  5. Authors should perform physicochemical characterization of DOX loaded decorated NPs under physiological conditions overtime. That data will be able to point out whether or not aggregation of particles is induced due to presence of serum and if that potentially induces cytotoxicity.
  6. Authors must highlight the significance of their coupling strategy in the conclusion. As per literature, there are hundreds of methods including a simple hydrophobic non- covalent binding of antibody to the nanoparticles. What makes this method so special? This part is not clear. 
  7. Authors have used the term "High Targeting efficiency" in the title while have tried to demonstrate " specificity" of the their nanoparticle constructs. While these two terms are similar, they have very different significance in context of drug delivery. In the current manuscript, authors haven't explored the efficiency of targeting for their construct but have pointed to their specificity. Authors should consider reflecting this in the title to avoid reader confusion.

Author Response

(The authors gave the same response as above.)

Round 2

Reviewer 1 Report

Gretat job! Congrarulation!

Author Response

We thank reviewer 1 for having no additional remarks 

Reviewer 2 Report

The revised manuscript is of interest to the readership and the improvements will help strengthen the manuscript.  A few minor points to address before publication.

For the amount of DBCO per mAb, the reviewer appreciates all the work and care in calculating amount of labeling of the mAb. The experimental conditions refer to 5 equivalent of NHS DBCO ester and the final results per UV describe 10 DBCO which does not fit the mass balance.  Please double check equivalents of NHS ester added or the UV/Vis calculations.  

  A stylistic point but the term anti-cancer is used in this text which is some  publications are used only for compounds that have shown activity in humans and the term cytotoxic antiproliferative should be used for compounds used in vitro and antitumor for compounds that have shown activity in vivo. Please check with MDPI or other scientific guidelines and general usage of the terms relating to cytotoxic agents and revise text accordingly. 

Below is a guideline from the journal of natural product from the American Chemical Society describing the difference: 

 Compounds that suppress the growth of, or kill, isolated tumor cell lines grown in culture should be referred to as either “cytostatic” or “cytotoxic”, as appropriate. Only compounds that inhibit the growth of tumors in animal-based models should be called “antitumor”. The term “anticancer” should be reserved for compounds that show specific activity in human-based clinical studies (see Suffness, M.; Douros, J. J. NatProd198245, 1–14).

Author Response

The response to reviewer 2 can be found in attachment 

Reviewer 3 Report

Zundert et al. have made significant edits and have satisfactorily addressed reviewers' comments. The authors have submitted an improved version of the manuscript, which would be deemed suitable for publication, following a minor comment to be addressed:

  1. Following up the discussion on usage of the terms  "targeting efficiency" and "specificity", the former is used when length-scale of targeting is on the order of organs, following in-vivo administration, while the latter is used more the length-scale is on the order of molecular level. As the studies in the current study are in-vitro the terms could be use interchangeably. However, a brief statement  defining what efficiency/ specificity means, especially in this context, would help avoid any confusion for the readers.

Author Response

The response to reviewer 3 can be found in attachment 
